# Lineages of Fractal Genera Comprise the 88-Million-Year Steel Evolutionary Spine of the Ecosphere

**DOI:** 10.3390/plants13111559

**Published:** 2024-06-05

**Authors:** Richard H. Zander

**Affiliations:** Missouri Botanical Garden, 4344 Shaw Blvd., St. Louis, MI 63110, USA; rzander@mobot.org

**Keywords:** adaption, complexity theory, extinction, fractal evolution, minimally monophyletic genus, phylogeny, Streptotrichaceae, tadpole lineage

## Abstract

Fractal evolution is apparently effective in selectively preserving environmentally resilient traits for more than 80 million years in Streptotrichaceae (Bryophyta). An analysis simulated maximum destruction of ancestral traits in that large lineage. The constraints enforced were the preservation of newest ancestral traits, and all immediate descendant species obtained different new traits. Maximum character state changes in ancestral traits were 16 percent of all possible traits in any one sub-lineage, or 73 percent total of the entire lineage. Results showed, however, that only four ancestral traits were permanently eliminated in any one lineage or sub-lineage. A lineage maintains maximum biodiversity of temporally and regionally survival-effective traits at minimum expense to resilience across a geologic time of 88 million years for the group studied. Similar processes generating an extant punctuated equilibrium as bursts of about four descendants per genus and one genus per 1–2 epochs are possible in other living groups given similar emergent processes. The mechanism is considered complexity-related, the lineage being a self-organized emergent phenomenon strongly maintained in the ecosphere by natural selection on fractal genera.

“Nature uses only the longest threads to weave her patterns,

so each small piece of her fabric reveals the organization of the entire tapestry.”

—Richard P. Feynman

## 1. Introduction

In several recent works [1,2], the concept of genus was redefined in terms of complexity analysis, that is, of emergent phenomena and the stabilizing features of fractal evolution. In the present paper, the lineage is redefined in the same context, as parallel branching series of minimally monophyletic groups, each lineage as a whole preserving resilient traits across more than 80 million years of environmental perturbation.

The modern bryoflora first appeared in mid to late Cretaceous times [3,4,5], about 80–100 mya. It is called “modern” because genera and species recognizable as fossils are still extant. How have these lineages survived such a long time? It may be postulated that natural selection encourages traits that stabilize floras through long-term environmental perturbations. The modern flora has, in the past 80 my, survived, at least partially, such environmental crises as extremes of global temperature, boloid impacts, supervolcanoes, glaciations, inundations, Milankovitch events, and orogenies. Is there evidence that extant species include traits making them resistant and resilient [6] to major environmental changes? Why are those traits not overwritten with each speciation event?

For investigation, we can use the minimally monophyletic genus, which has several valuable properties. The *microgenus* is here defined as the smallest unit of monophyly based on taxonomically important expressed traits. As the basic unit of evolution [2], it clearly represents empirically based emergent phenomena, i.e., taxa higher than the species rank, in analysis, and which are not the immediate result of a clearly defined process other than natural selection. Possible applications of microgenus-associated complexity-based processes in nature are fractal and self-similar, thus widely applicable [7]. Self-similar features of stabilizing evolution simplify prognostication of floral futures [8,9] in an increasingly chaotic biosphere [2]. The microgenus has been statistically well-supported by conjugate priors in past Shannon–Turing analyses [2,10] based on data from more than 30 microgenera. In contrast, the minimally monophyletic cladistic group is two sister taxa, i.e., two taxa whose traits imply they closely share one ancestor. This has no direct information about their possible ancestor–descendant relationship.

The microgenus enables certain forms of evolutionary, ecological, and geo-historical analysis heretofore difficult and problematic. Genera are commonly envisioned in the literature [11] as polythetic, with more than one ancestral species and overlapping diagnoses for the species. These may be called *mesogenera,* which are convenient for sorting and simplifying the classification of species. Given the cladistic principle of strict monophyly, species are often clumped into large groups in molecular studies, or *macrogenera*. Both cladistic mesogenera and macrogenera fail to recognize the ancestor–descendant processes attendant on the production of stable, resilient evolutionary groups. Although Van Valen [12] earlier pointed out that ancestral species were both extant and common, there are about half the terminal species in a cladogram that are actually ancestral to another species [2], and ancestral species are usually not identified as such in cladistic study [13,14,15,16]. This results in loss of important information for the inference of evolutionary processes.

The genus, here the minimally monophyletic group informally termed the microgenus, has been demonstrated (primarily [2], but also [1,17,18,19,20,21,22,23,24]) to be optimally of one ancestral species and four immediate descendant species. It thus has a fractal dimension of 1.16 (log5/log4), being self-similar across scales, including the next higher taxonomic rank (the supergenus or subtribe, for an example, see *Neotrichostomum* as the super ancestor of four other genera [2]). It has, optimally, four new traits in both ancestral species and descendants, which is characteristic of the Pareto distribution [25] of 20:80, which likewise has a fractal dimension of 1.16. Microgenera may be considered phenomena that are emergent out of a chaos ruled in part by natural selection. Critical similarities across genera include the fact that the new morphological traits of the ancestor passed without change to all optimally four descendants, resulting in new species well-equipped for sympatric survival. As such, NK complexity analysis using random Boolean networks [2,26,27] finds that this may reduce competition between descendants in any environment.

Natural selection is the foremost complexity-related process generating taxa as self-organized, emergent phenomena contributing to biological diversity. In this paper, we examine the possible selective value of fractal evolution affecting lineages of species, with a taxonomic group of the author’s specialization, the family Streptotrichaceae of the mosses, as the test subject.

## 2. Results

### 2.1. Calibration of a Single Microgenus

Calibration estimates the beginning of a lineage or group of related lineages. The existence of the West Indies archipelago is comparatively limited in geologic time and serves to isolate single non-concatenated microgenera, in this case *Chionoloma* and *Tainoa,* each of five species [2]. If we take 45 my as the maximum time depth of the probability box and 3 mya at the end of Miocene as minimum and use a Fermi estimation of the geometric mean as an estimate of generation of endemic microgenera in West Indies, then, the geometric mean of the range 3 to 45 million years is the square root of 3 × 45, or 135, or 11.6. Thus, the Fermi estimated time of generation of, say, the West Indian, mostly endemic, genera *Chionoloma* and *Tainoa* is 45 minus 11.6, or 33.4 mya, or early Oligocene. 

Another way of estimating the generation of genera native to the West Indies is to assume a normal distribution of the data of genus generation and assume that the range of 45 to 3 my is covered by four standard deviations. We then discard the fourth standard deviation. Dividing the range of 45 minus 3 my by 4 gives 10.5 my. Subtracting 10.5 my from 45 my yields 34.5 mya for the probable generation of *Chilonoloma* and *Tainoa*, which is a close match. In support, Ricklefs and Bermingham [28] estimated the average persistence time of all studied lineages in the West Indies as 34.5 my, a surprising match.

Thus, we may estimate that about 34.5 my is a standard range of time for generation of a new genus, occurring within two standard deviations around the probabilistic mean, that is, 45 divided by 2 or about 22 my. This is obviously speculative, but it is a first pass at developing an explanatory theory. The complexity analysis of evolution is, after all, a rather new field. A positive feature of complexity analysis is the simplification of the evaluation of emergent and fractal phenomena due to apparently similar underlying generative processes and self-similarity across scales.

### 2.2. Calibration of Lineages

Assuming the generation of one genus from an ancestral species in another genus takes 22 ± 17.25 my on average, and assuming a punctuated equilibrium burst of descendant generation, then the number of ancestor to ancestor steps measures the temporal depth of a lineage. In the caulogram of Streptotrichaceae [1], there are four steps (Figure 1 and Figure 2) along the most lengthy lineage from one ancestor to the next; thus, the family might be four times 22 my in age or 88 my. The late Cretaceous era begins ca. 100 mya, and the K-T boundary is at 66 mya. 

This admittedly coarse estimate for a lineage of modern moss taxa is nevertheless in line with estimates of the beginnings of the evolution of the modern moss flora. According to Bechteler et al. [3], extant bryophyte families diversified across the Cretaceous and early Cenozoic, with the majority occurring during the Cretaceous terrestrial revolution, while climate fluctuations [29,30] during this time encouraged the diversification of bryophytes in arid (Pottiaceae) and exposed (Funariaceae) lands. Jauregui-Lazo et al. [31] reported in a molecular study the age of the *Syntrichia* (Pottiaceae) lineage as 50.1 ± 6.3 mya (early Eocene) with a recent diversification at 15–12 mya, and they further estimated that Streptotrichaceae (as *Leptodontium capituligerum* and *L. pungens*) broke from the stem Pottiaceae at about 120 mya. The minimum stem age of the Rabdoweisiacese in their calibration is 66–84 mya (also Late Cretaceous). Villareal et al. [32], in another molecular study, found that liverworts diversified about the same time, although the crown group was tracked much earlier.

If Jauregui-Lazo et al. [31] are correct in estimating the origin of the Strepotrichaceae at 120 mya, this provides a stretch of 32 my for the evolution and extinction of the ultimate predecessors of extant taxa of Streptotrichaceae. The present estimate of 88 mya for the beginnings of the Streptotrichaceae also implies that the Streptotrichaceae lineage endured and survived all the global environmental perturbations between then and the present, including the K-T and early Eocene temperature maxima, etc. The earliest lineages, those with only one step from the basal taxon, are the unispeciate genera *Leptodontiella* and *Trachyodontium* and the generitype of the rather evolutionarily dissected genus *Williamsiella* [2], all of the Andean range occurring on twigs and branches, and the unispeciate *Austroleptodontium* of New Zealand on sand and lawns. 

The full phylogeny is given here (Figure 1) with genera in different colors, all taxa named, and numbers of traits in the novon (new descendant character states as changed from old ancestral states). The Streptotrichaceae has 10 microgenera, totaling 30 species. The total number of direct descendants in each of the four temporal categories above is 4, 8, 3, and 0, with an average of 3.75. The average number of species per microgenus is then 4.75. 

The phylogeny (Figure 1) is somewhat in the outline of a tadpole, with maximum taxa and maximum evolutionary activity in the most recent intervals of 22 my (Figure 2). We may speculate that natural selection at the ecosystem level [33] has maintained this string of ancestral traits since the late Cretaceous. Below are listed the genera for each stretch of time, numbers of species (ancestor plus immediate descendants), and numbers of secondary descendants separated by colons; this is from [1] (p. 72). The code for species is x (y:z), where x = total species in genus, y = ancestor plus direct descendants, and z = descendants from secondary ancestry (descendants generate their own descendants). 

Secondary ancestry is here conceived as due to allopatric dispersal to somewhat different habitats not requiring the preservation of the full immediate ancestron. Total numbers of species in the family originating in the 22 my intervals estimated in the present study are reduced, presumably by extinction, over time, that is, to 12, 9, 8, 1. Also, ancestors plus only immediate descendants are reduced to 6, 9, 6, 1. Secondary ancestry is then common only in the most recent interval, with 6 secondary descendants. 

2–22 mya Miocene and beginning Pliocene. 

***Microleptodontium***, 5 (2:3) ex *Leptodontiella* via shared ancestor with *Rubroleptodontium****Stephanoleptodontium***. 7 (4:3) ex *Williamsiella*

22–44 mya mid-Eocene and Oligocene. 

***Crassileptodontium***, 4 (2:2) ex *Trichyodontium****Leptodontium***, 4 (4:0) ex *Williamsiella****Rubroleptodontium*** 1 (1:0) ex *Microleptodontium* unknown ancestor

44–66 mya Paleocene and early Eocene. 

***Austroleptodontium***, 1 (1:0) ex *Streptotrichum****Leptodontiella***, 1 (1:0) ex *Streptotrichum****Trachyodontium***. 2 (2:0) ex Streptotrichaceae Ur group***Williamsiella***, 4 (2:1:1) ex *Streptotrichum*

66–88 late Cretaceous. 

***Streptotrichum***, 1 (1:0) and **Streptotrichaceae Ur group**, ex early Pottiaceae.

Jauregui-Lazo et al. [31] produced a massive global study of the pottiaceous genus *Syntrichia* (Pottiaceae), mapping that genus and relatives against a linear scale of geological time, invoking a strict molecular clock. In groups *Aesiotortula* and “NHR” [31], it is obvious that the molecular species cluster in small groups. If we take these as representing the descendants of microgenera, then, the average number of descendants in both large groups is 3.5 for *Aesiotortula* and also 3.5 for “NHR”. In a study of West Indian Pottiaceae [2] of eight microgenera including 30 species, each genus had an average of 3.75 species. Even without actual analysis of *Syntrichia* morphology, one might expect to find microgenera in that macrogenus similar in composition to those in Streptotrichaceae [1] and the West Indian study [2].

### 2.3. Idealized Maximum Pressure on Reserve Ancestron

The portion of ancestral traits that are not used in generating new traits on an immediate descendant species is the reserve ancestron. Each descendant species of a microgenus ancestor has about four new traits that represent state changes in the reserve ancestron, that is, the descendant species now lacks that ancient character state because it is used up in generating a new state adaptive to the present environment. How much is the store of traits adaptive to ancient environments depleted in a lineage since the origin of the modern moss flora in the late Cretaceous? We test this by subjecting a known evolutionary lineage, the Streptotrichaceae, to maximum character state change in the reserve ancestron as constrained by (1) recent traits of an ancestor are unchanged in a descendant species, and (2) new traits of descendant species are each different in a genus, which are features of fractal evolution previously established [2].

Figure 2 summarizes idealized maximum pressure on the reserve ancestron. Species (here unnamed) and genera of the Streptotrichaceae (Bryophyta) are arranged as previously given [1]. The above estimated dates of origin of the genera are assigned geologic time periods, largely at 22 my intervals representing a strict morphological clock (a simple linear scale). Species are assigned alphabetic letters with each letter representing the average of four new traits in a descendant species as previously determined for this family (actually 4.04). Traits accumulate in a species at the rate of one letter (4 traits) per speciation event, which simulates four state changes from ancestral adaptive states to a newly adaptive states. 

There are 171 character states possible in the Pottiaceae and its segregate Streptotrichaceae [34], including both two-state and multistate characters. The total number of character states represented in the caulogram (Figure 2) is 124, or 73 percent. This represents a change in 124 character states in the reserve ancestron, a large percentage. On the other hand, as a whole, the extant species of the Streptotrichaceae have lost only four character states from the joint reserve ancestron, that is, the letter A occurs in all species. Each sub-lineage loses only the number of ancestron traits represented by the letters shared by all species in it, and the same is true for each genus. 

The analysis is an evaluation of maximum pressure on the reserve ancestron of one family. The actual numbers of new traits per species (per speciation event) averages 4.04 but may range from 3 to 5.75 average traits per species in each genus (with a high of 11 in *Williamsilella auraucarieti*) [1,2]. A breakdown of the geographic range and habitat is given by Zander [1] (p. 201), which is relevant to the analysis of genus origin in geologic time but is not the subject of the present paper.

## 3. Discussion

Previous papers [1,2,10] have emphasized the evolutionary importance of the novon of new traits and the immediate ancestron of shared new traits of the ancestral species. Here, it is conceived that the narrow lines in dendrograms connecting species and genera (e.g., Figure 1 and Figure 2) represent all the traits of a species that are apparently not critical for survival in the present regional or local environment. One may speculate, however, that relic traits are important to the ecosystem as having been active in survival during environmental perturbations of the geologic past. As subordinate character states, they are the source of all new character states in newly evolved descendant species. Such highly adaptive but presently inactive traits, in what we might call the *reserve ancestron*, are not used up lightly. 

All lineages of modern bryoflora that are more than four microgenera in depth probably reach back in time to the Cretaceous–Paleocene boundary of 66 mya or beyond. During the past 100 my, lineages with extant species have adapted to major environmental perturbations [29,30] by physiological and morphological adaptations, migrating, switching substrates, and other strategies now in part engraved in their ancestrons. Temperatures today average 14–16 °C, with diurnal variation of about 15 °C, ranging from 8 °C (temperate cities, tropical forests) to 40 °C (highland deserts), while tropical areas today range from 18 to 30 °C [35] (p. 91). Tropical North Atlantic temperatures averaged around 30 °C ca. 90–100 mya during the Late Cretaceous Thermal Maximum. Tasmania in the Eocene was equally warm, up to 30 °C [35] (p. 110), reflecting the Paleocene-Eocene Thermal Maximum, with global temperatures about 14 °C higher than the present 1900–1990 average. Global temperatures slid downwards during the mid and late Cenozoic (Oligocene and Miocene) with the polar ice sheets becoming established about 34 mya at the beginning of the Oligocene [36], with sea levels about 125 m lower than at present, during the glacial maximum about 25 kya. 

There have been several large igneous eruptions (Deccan, North Atlantic, Ethiopia and Yemen, Columbia River) associated with extinction events [37,38,39] since the Cretaceous, plus a myriad identified smaller yet quite destructive sites [40]. Super-volcanoes are related eruptive events but are more limited in area, although regionally destructive by sulfuric acid emissions and physical force; within the past 100 my, there have been 15 well-documented super-volcanoes [41,42]. Boloid impacts [43] causing major extinctions are few, for example, the Caribbean Chicxulub impactor at the Cretaceous-Paleogene boundary 66 mya, but smaller impacts that may cause the equivalent of a nuclear winter occur about two or three times every million years [44,45]).

With global cooling during the Oligocene, Miocene, and Pliocene, Milankovitch events [46,47] triggered a series of Pleistocene glaciations [48]. Milankovitch events are large changes (swings of up to 8 °C) in the Earth’s climate caused by obliquity in the planet’s axis, eccentricity of the orbit, and precession of the axis; events occur every 110,000, 41,000 and 28,000 years and happen whatever the global temperature is and, thus, affect biodiversity back into the Cretaceous. There have been several major events in the past 100 ky, including the Laschamps excursion of 41 Kya lasting 1.8 ky, the Norwegian-Greenland Sea excursion of two minima at 65 and 60 mya, and the Mono Lake/Auckland set of excursions of 34 and 29 kya lasting [49,50]. 

There is much ado in the literature about extinctions, sometimes massive, associated with environmental shocks such as those in the partial summary above. The process ensuring tolerance of a lineage to a circa 60 my set of perturbations may be revealed in a complexity analysis of the intra- and interrelationships of microgenera and their extended lineages. This paper attempts to demonstrate that modern lineages have endured at least the major disruption at the Cretaceous-Paleogene boundary (66 mya) consisting of the double punch of a very large boloid impact in the Caribbean and the large igneous eruption of the Deccan Traps. If the Streptotrichaceae began with *Streptotrichum* as a five-species genus in the late Cretaceous, it has then generated nine additional genera. This presumes an optimal four descendant species plus ancestor in each. The total extant species is 30. After the extinction of two-fifths of the inferred potential species (five species times 10 genera), the family is robust and is represented today by 30 species. This is not a bad record after 66 my of environmental pruning, although it may be stipulated that many species occur in montane, arboreal refuges [1] (p. 203). 

If the branching order of evolution were indeed dichotomous, as in a cladogram, each speciation event would eliminate, by state changes, optimally four traits per event from the post-adaptational reserve ancestron of the lineage. The turn-over of reserve traits is, however, different with microgenera, in that the microgenera lineage links series of ancestral species, creating a burst of speciation (four descendants) all based on a single shared immediate ancestron. In this case, the new traits of the descendants are sampled from the reserve traits, but all new traits are different in each immediate descendant. Given that cladistic analysis does not identify ancestral species when extant, much information on the mechanics of lineage evolution is lost. The saturation of newly adaptive species of an ecosystem through speciation with microgenera is more rapid than if dichotomously split. The former multiplies by four for each speciation event, while cladistic speciation multiplies by two per event.

Because each of the (average) four descendant species has different new traits, the whole set of descendants in any one genus represents no change in those ancient traits. Ancient traits are indeed lost at the beginning of a lineage, however, but only for that lineage. Inasmuch as there are maximally only four concatenated microgenera in a lineage, however, one must assume that the loss of 16 traits in the reserve ancestron is made up for by lineage branching (parallel lineage sorting) and by gradual elimination of genuinely non-adaptive traits out of the ca. 171 character states attributed to morphological study of the Pottiaceae [34] including its close segregate, the Streptotrichaceae. The lineages across 88 my are then conservative of adaptive traits and stable. If complexity processes are general in appearance and self-similar across scale and taxon, then the skein of lineages parallel across more than 88 my of time for the modern flora is a steel backbone for resilience in ecosphere evolution. 

Caulograms of West Indian genera [2] (p. 34, 41) show clearly bursts of speciation apparently still intact after millions of years of existence in this island archipelago [28]. The caulograms are only one or two linked ancestral species in depth, implying that recent geographically restricted genera have changed little since derivation from a globally widespread ultimate ancestor (*Neotrichostomum*) at the caulogram base. One might theorize that microgenera are frozen in stasis after an initial burst of speciation, which would be a contemporary equivalent to fossil-based punctuated equilibrium. Survival of the descendants as a lineage of genera in stasis ensures that at least a portion of the reserve ancestron long remains. It is the lineage of microgenera about four genera deep that is the major operator of evolution over geologic time. 

Given the persistence of modern moss taxa through the vicissitudes of 88 my of environmental perturbation, one must seriously consider the mechanics suggested above of storing selected adaptive traits for epochs as an ancestral reserve of genomic memory. This may be visualized as an informative evolutionary spine stabilizing ecosystems. Given that these are self-organized, emergent phenomena self-similar across scale, the process may well be universal in the ecosphere.

The word “invasive” in the context of 88 million years of global adaptive evolution of individual lineages becomes less meaningful. The extant, surviving lineages provide maximum control over energy flows [51], because this is a result of natural selection at the lineage and ecosystem level. I think great biodiversity supports the largest stable biomass or coverage over the vicissitudes of time. The only really destructive invasive organism is humankind because we are game-changers. In all other cases of apparent invasion, one can consider that, over time, invaders are either replacement species or healing species. It is presumptuously self-centered to label species that encroach on already human-damaged environments as somehow negative and to be extirpated. The fundamental goal of natural selection is entropic health through proper levels of interactive, genetically well-prepared biodiversity well connected and stable through space and time. 

## 4. Materials and Methods

The calibration of microgenus emergence times was estimated for two not-concatenated microgenera, *Chionoloma* and *Tainoa* [2], and for an entire branching lineage, Streptotrichaceae [1]. A strict morphological evolutionary clock was used, methodologically parallel with the strict molecular rates used in phylogenetic studies [31,32] dating taxon emergence in deep time.

Analytic methods used for the delimitation of microgenera are fully discussed by Zander [1,2]. In sum, the single ancestral species was identified morphologically as that species most similar to an outgroup and also least advanced compared to other ingroup species [2]. The characters used as the sources of new traits (character state changes) in descendants are apparently those older and, for the present time, less important traits in the trail of characters of the ancestral species. Given that older traits are the sources of resilience to environmental perturbations similar to those events survived in the past, speciation uses up this resource for immediate survival. Living fossils strongly demonstrate the value of stasis associated with evolution preserving ancient well-tested traits.

Riclefs and Berminham [28] found that rates of extinction in the West Indies are low, suggesting that taxa remain stable in island areas in part due to isolation. Persistence of lineages was found to be long, anywhere from 2 million years (birds in Lesser Antillies) to 34 million years (reptiles in Greater Antillies) depending on the organism, with an average persistence of 34.5 my. 

In the case of the “Weissia Probe” of the West Indies Pottiaceae mosses [2], the two largely endemic genera, *Chionoloma* and *Tainoa*, were largely intact, mostly with four descendant species and the genetic distance in terms of novon traits between each genus and the basal globally distributed ancestral species little different than that between the descendant species and their immediate ancestor. 

From this, one might conclude that there are no extinct genera between the extant microgenera and their shared globally distributed immediate ancestor, *Neotrichostomum crispulum*. It is safe to say that we can assume that this single step in generating each genus separately took place sometime in the past 30–40 million years. 

Such a calibration is a *probability box*, wherein we know the time-wise bounds but little about the distribution of data in the box, most particularly, when was the ancestral species of a genus generated. Similar questions have to do with the geographic distribution of the species and possible secondary ancestry eventually leading to new genera. Orthogonal to the probability box might be devised graphs of environmental changes and perturbations that may aid in the inference of speciation events. Certainly, the order of speciation of the ancestral species provides information on the depth of time that extant genera have lasted to the present.

Given that the actual presence of a particular substrate or habitat is necessary for the survival of a species, then, the date of emergence of islands or continental land masses or portions of them is the most distant bound in terms of millions of years ago. The West Indies emerged from the sea or was cut off from the mainland. 

The Lesser Antilles originated about 20 mya, arising from a subduction zone [52,53]. The Greater Antilles are more complex in geological history, and portions of the land masses may have remained above sea level since the mid-Eocene (45 mya) excepting periodic inundations of portions of the area. 

The Streptotrichaceae is the only family of mosses to date that has been taxonomically reduced to microgenera, these being one ancestral species and a few immediate descendants with occasional secondary descendants. As such, the caulogram (Figure 1 and Figure 2) shows four levels of microgenera in depth. This implies, by the West Indian calibration and a strict morphological clock, that it is four times 22 my in age, or 88 my, reaching back to the late Cretaceous. Supportive of this is the calibration of the Jauregui-Lazo et al. [31] study establishing a fossil haplolepidous species (*Cynodontium luthii*, Rhabdoweisiaceae) at between 66 and 84 mya, with Streptotrichaceae branching off the Pottiaceae earlier at ca. 115 mya. 

The morphological traits are given in reference [2]. These traits are not used to calibrate the morphological clock. Two separate genera derived from a closely shared ancestor were isolated in the recently emerged West Indies. These calibrate the unit of 22 my for the clock, and at four genera deep in time, number of units reach 88 mya. This initial start for the lineage, and the equal time assigned the intervals, is further implied by the beginning of the modern moss flora near the end of the Cretaceous. A third calibration is the split of Streptotricaceae from Pottiaceae of 120 mya estimated by a molecular clock [31]. The number of morphological traits is the (short) distance between ancestors and ancestors and between ancestors and descendants that supports the coherence of the lineage as a whole.

## 5. Conclusions

Traits best for the ecosystem are not necessarily those best for the species. There are ca. 171 morphological character states available in Pottiaceae and Streptotrichaceae. Of these, about 124 are in play in Streptotrichaceae at any one time. Only a few of these are demonstrably (by existence of the species) valuable at any one time. The remainder are presently apparently unused or of little use. The novons of the ancestor and of the descendant species are effective in adaptation to present environmental conditions, the novon of the ancestor for sympatric survival, the novons of the descendants for peripatric survival, and allopatric insertion into entropic niches. 

A key fact is that the immediate ancestron is present in all immediate descendants of a microgenus ancestor species and is fairly stable during secondary ancestry. I posit that the presently unused traits of the ancestron, here termed *reserve ancestron*, are strongly conserved features of adaptations to past environmental perturbations and are valuable to the ecosystem in the long term, geologically speaking. We may consider *Crassileptodontium* (Figure 1 and Figure 2)*,* because it has the ancient *Trachyodontium* as the immediate ancestral genus, an example of how ancestral taxa may be a source of valuable adaptive traits.

Preservation of these keys to past survival is effectuated in the following manner. The immediate ancestron doles out an optimal new four traits to each of optimally four descendants during a speciation burst, with each descendant receiving four traits different from the sets given to the other three descendant species. Each new trait is a state change in one of the traits of the reserve ancestron. In this way, of the 171 character states available, only four traits of past value are put at risk at a time if a descendant goes extinct. This preserves 96 percent of the reserve ancestron per burst of speciation. Given that microgenera are fractal at dimension 1.16, with “genusation” resulting in one ancestral taxon and optimally four descendants, each microgenus presents its newly inhabited 1–2 epochs with about 16 new traits in containers preserving the four most recent adaptive traits (the immediate ancestron). This optimality may well be the selective driver for extant punctuated equilibrium. In any case, the continued presence of a lineage with genera reaching back to the late Cretaceous means that almost all traits are preserved in that lineage. 

No explanation is available for why there are optimally four traits per species, four descendant species per ancestor, and up-the-scale four ancestors for each penultimate ancestor. A similar problem exists in material sciences where there is an anomalous abundance of inorganic compounds whose primitive unit cell contains a number of atoms that is a multiple of four. This feature has been named the *rule of four* [54]. It seems to correlate with low symmetries and loosely packed arrangements maximizing the free volume, but no decisive explanation is evident. 

As a microgenus of one ancestor and optimally four descendants is an important element in evolutionary processes, molecular mutations of less than 22 my distance in time are not informative. Only the order of descent of the ancestral species of each genus is revelatory. Of course, true species are commonly distinguishable by distinctive molecular signatures, but molecular signatures cannot distinguish cladogram terminal groups with zero or one expressed traits from true species with two or more expressed traits following the evolutionary rule of four optimal traits.

A way to identify those traits that have been in the past valuable to the genus and ecosystem is to list the salient traits of microgenera with only one or two species, which are morphologically distant from related genera and, therefore, probably ancient rather than newly generated. A comparison of those traits with those of similarly partially extinct minimally monophyletic genera in the same environmental situation should identify at least a few similar traits that survived recent environmental perturbations. Those are the traits that are best preserved by conservationists, while the lineage as a whole works through complex processes to preserve itself. 

## Figures and Tables

**Figure 1 plants-13-01559-f001:**
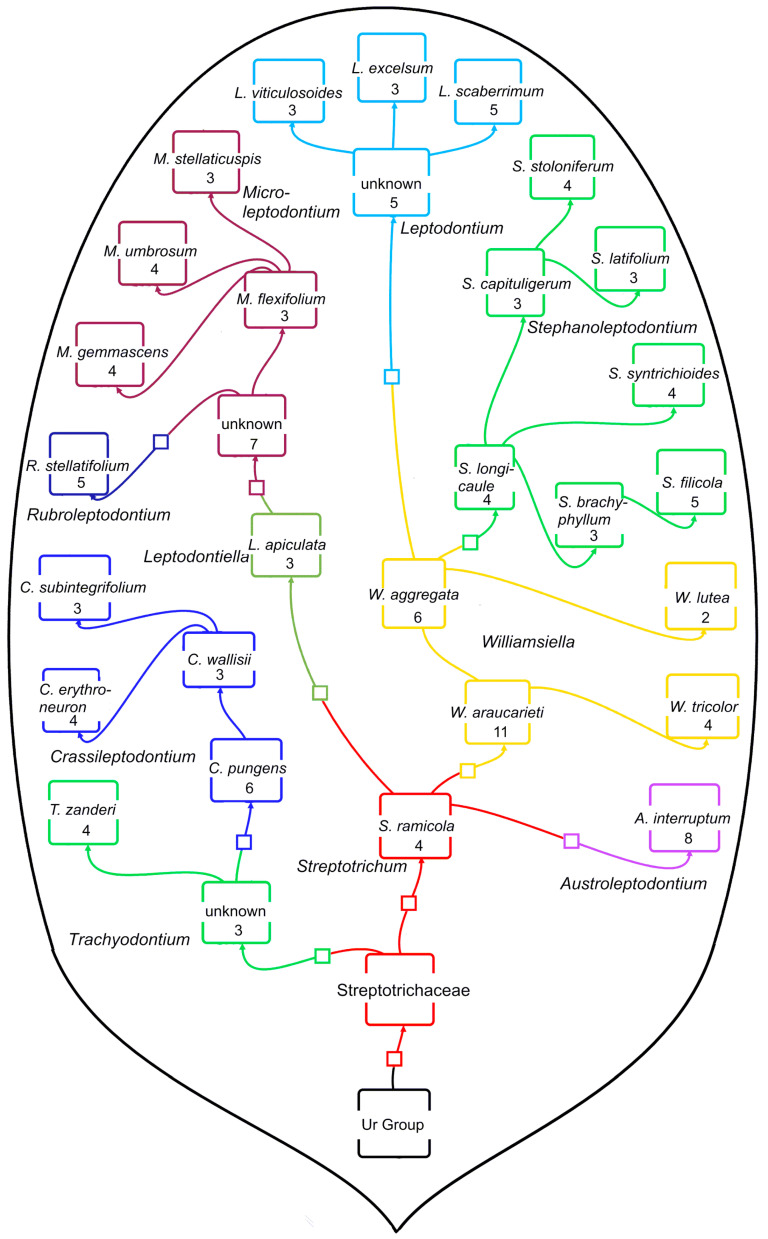
Streptotrichaceae lineage, showing estimated directions of species generation and numbers of traits in the novon (set of new traits). Genera are distinguished by color matched in Figure 2. Recent genera are more speciose; ancient genera are apparently trimmed by natural selection but not entirely extirpated from the ecosphere. This tadpole shape may be universal among long-lasting families because these give a survival advantage to an ecosystem over geologic time. The “Ur Group” is probably early elements of the closely related Pottiaceae.

**Figure 2 plants-13-01559-f002:**
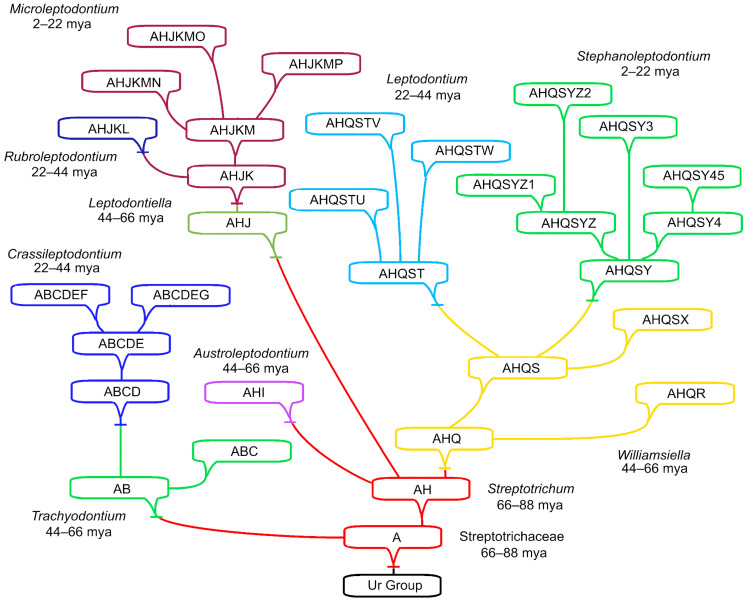
Idealized maximum pressure on reserve ancestron. Species (here unnamed) and genera of the Streptotrichaceae (Bryophyta) arranged in a previously reported caulogram. Estimated dates of origin are assigned geologic time intervals. Species are assigned letters each representing the average of 4 new traits in a descendant species previously determined for this family. Traits accumulate at the rate of one letter per speciation event, which simulates four state changes from ancestral adaptive states to four newly adaptive states. Constraints are (1) recent traits of an ancestor are unchanged in a descendant species, and (2) new traits of descendant species are different in a genus. Percentages are given for percent of character states changed in the reserve of anciently adaptive traits in the entire lineage and for each genus, although only four states are permanently lost for entire lineage and only four when limited to a sub-lineage.

## Data Availability

All data are provided in the present manuscript.

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
