# Peer review of "Lineages of Fractal Genera Comprise the 88-Million-Year Steel Evolutionary Spine of the Ecosphere"

_plants, 2024, doi:10.3390/plants13111559_

Round 1
Reviewer 1 Report
Comments and Suggestions for Authors
The paper presents very interesting features of originality, especially for the developments that they may have in the study of the phylogeny of bryophytes. We hope that such research can also be extended to other genera and species in order to have a broader framework of knowledge.
Reviewer 2 Report
Comments and Suggestions for Authors
plants-3008666
In this manuscript, a study on the possible selective value of fractal evolution affecting lineages of species, using as test subject the mosses taxonomic group of the family Streptotrichaceae, was carried out.
In this investigation the salient traits of microgenera with only one or two species have been considered, and which are morphologically distant from related genera and thus probably ancient rather than newly generated. Comparison of those traits with those of similarly partially extinct minimally monophyletic genera in the same environmental situation should identify at least a few similar traits that survived recent environmental perturbations, including effects related to climate change and to influences of the environment on biodiversity. This approach could give a view of the whole lineage, preserving itself and originated through complex processes.
The Materials and Methods section needs to be divided in paragraphs in order to give a clear description of the approach used in this study.
Revisions
lines 44-45: ‘it is clearly represents empirically based emergent phenomenon’, please clarify this sentence;
49: ‘hoas’ change to ‘has’;
line 70: ‘Neotricdhostomum’ change to ‘Neotrichostomum’;
line 144: ‘captituligerum’ change to ‘capituligerum’;
line 156: ‘Williamsiella’ please, give information and references related to this genus;
line 161: ‘from from’ delete repetition;
line 186: ‘Leptodntium’ change to ‘Leptodontium’;
line 200: ‘Asiotortula’ please, give information and references related to this term;
line 203: ‘Aesiortula’ please, give information and references related to this term;
line 240: ‘Williamsilella auraucarieti’ please, give information and references related to this species;
lines 398-399: ‘taxonomicaloly’ change to ‘taxonomically’;
line 404: ‘tuthii’ change to ‘luthii’;
line 445: ‘therefor’ change to ‘therefore’.
Reviewer 3 Report
Comments and Suggestions for Authors
This article is written in a style unusual for most current publications in botany.
The author, Richard H. Zander, revolts againt mechanic application of cladistic
methods without seeing anything behind series of divergences. His non-standard
macroevolutionary cholistic approach to things, which looks simple because of
over-simplifications, led him to an expanded involve into consideration many
general issues and philosophical discussions for their better understanding.
The reviewed short article of Zander continues the development of his
macroevolutionaly theory, described previously in two thick books (cited as [1] and
[2]). A lineage, usually classified in Pottiaceae, accepted here as Streptotrichaceae,
is in the focus of the study. Here the Streptotrichaceae is discussed as a model
object for various comparisons and estimates.
The above mentioned makes the text difficult to follow for readers who never
studied (and well understood) these [1] and [2]. However, it would be certainly
impossible the embed even the main ideas from these books to the introduction of a
journal article, as this will inflate its volume too much. However, the author has to
consider the readers who never read [1] and [2].
The datamatrix of morphological traits shown in Figs. 1-2 and used for a strict
morphological evolutionary clock has to be available for the readers (at least as
Supplementary materials, but better in the paper text). I'd be insistive with this, as
the counting of morphological tratis for the calibration by morphological clocks is
not as straighforward as counting of ATGC substitutions. Also, the statement from
the Abstract "Analysis simulated maximum destruction of ancestral traits in that
large lineage" can't be understood without the list of traits.
There are some disputable assumptions, e.g. that the equal time (22 mya) needed
for the microgenera origination. However, the paper, first of all, have to be
improved for ultimate clearness of Fig. 1 and 2, for their contents and methods of
their building.
There are some typos and inexact statements:
line 47
[70. Self-similar must be line [7]. Self-similar
line 94
33.4 mya, or mid Oligocene - !Oligocene is accepted now on 33.9-23.0 mya
line 116
Late Cretaceous begins 70 mya - cf. line 146 84-66 mya (also Late Cretaceaous)
line 282 There are periods when the geomagnetic field is lower and multipolar,
destroying protection from external radiation, increasing cosmic ray flux to 2 to 3
times that at present. - What's the relevance to Streptotrichaceae?
line 355
Calibration of microgenus emergence times was estimated .... for an entire
branching lineage, Streptotrichaceae [1].
I found nothing about the microgenus emergence calibration in [1]; it would be
helpful to known the page number.
line 515
reference number 34 is used twice to
Zander, R.H. 1993 and Zachos 2001
Round 2
Reviewer 3 Report
Comments and Suggestions for Authors
with the given corrections, the msc is acceptable.